# Antibiotic Resistance Properties among *Pseudomonas* spp. Associated with Salmon Processing Environments

**DOI:** 10.3390/microorganisms10071420

**Published:** 2022-07-14

**Authors:** Gunn Merethe Bjørge Thomassen, Thorben Reiche, Christine Eikås Tennfjord, Lisbeth Mehli

**Affiliations:** Department of Biotechnology and Food Science, Norwegian University of Science and Technology (NTNU), 7491 Trondheim, Norway; thorben.reiche@ntnu.no (T.R.); chrisete@stud.ntnu.no (C.E.T.)

**Keywords:** antibiotic resistance, food processing environment, *Pseudomonas*, WGS, aquaculture

## Abstract

Continuous monitoring of antimicrobial resistance in bacteria along the food chain is crucial for the assessment of human health risks. Uncritical use of antibiotics in farming over years can be one of the main reasons for increased antibiotic resistance in bacteria. In this study, we aimed to classify 222 presumptive *Pseudomonas* isolates originating from a salmon processing environment, and to examine the phenotypic and genotypic antibiotic resistance profiles of these isolates. Of all the analyzed isolates 68% belonged to *Pseudomonas*, and the most abundant species were *Pseudomonas fluorescens*, *Pseudomonas azotoformans*, *Pseudomonas gessardii*, *Pseudomonas libanesis*, *Pseudomonas lundensis*, *Pseudomonas cedrina* and *Pseudomonas extremaustralis* based on sequencing of the *rpoD* gene. As many as 27% of *Pseudomonas* isolates could not be classified to species level. Phenotypic susceptibility analysis by disc diffusion method revealed a high level of resistance towards the antibiotics ampicillin, amoxicillin, cefotaxime, ceftriaxone, imipenem, and the fish farming relevant antibiotics florfenicol and oxolinic acid among the *Pseudomonas* isolates. Whole genome sequencing and subsequent analysis of AMR determinants by ResFinder and CARD revealed that no isolates harbored any acquired resistance determinants, but all isolates carried variants of genes known from *P. aeruginosa* to be involved in multidrug efflux pump systems.

## 1. Introduction

Antimicrobial resistance (AMR) is one of the major public health challenges of the 21st century [1,2,3,4]. The emergence of AMR has led to the ineffectiveness of common antibiotics and an increasing failure rate in treatment of infections, resulting in rising mortality rates for common infectious diseases [2,5]. One of the key driving forces in this evolving problem is the extensive use and misuse of antimicrobial agents [3,6]. Global antimicrobial use in aquaculture is expected to increase 33% between 2017 and 2030 [7]. The most widely used antibiotics in aquaculture, globally, belong to the antibiotic classes of quinolones (oxolinic acid, flumequine, and enrofloxacin), tetracyclines (oxytetracycline), amphenicols (florfenicol), and sulfonamides [7,8,9]. In the two largest salmon producers, Norway and Chile, florfenicol and oxolinic acid are most used [8,9].

The primary purpose of antimicrobials is to kill or inhibit the growth of microorganisms [10]. Bacteria present in the food value chain are affected by antibiotic treatment of livestock and farmed animals as well as by disinfection and sanitation agents used in the farming and food processing environment. The strong selective pressure this puts on bacteria promotes the development of tolerance and resistance properties [2,11]. The food chain contributes to the transmission of AMR through contamination of food products by resistant bacteria at different stages in the value chain and thereby functions as a vehicle for AMR dissemination [6,12]. Consequently, the food value chain may expose humans to antimicrobial resistant bacteria [13].

The contribution to AMR in aquaculture is not well understood [14]. Aquaculture is a highly complex and dynamic system influenced by different factors including both environmental and biological factors [14]. The system naturally contains high numbers of diverse bacteria [15] and bacteria with AMR properties are moved down the seafood value chain and can promote dissemination of AMR genes [16].

The main focus of AMR research and surveillance is on clinical isolates and a few indicator bacteria. There are very few studies investigating the occurrence of AMR among other bacterial species in food and in the food value chain in Norway. However, in a recent study concerning antibiotic resistance in *Pseudomonas* spp. from Norwegian chicken meat, they found that 21% of the *Pseudomonas* isolates (excluding isolates from agar with ciprofloxacin) showed resistance to more than three antibiotics and a high number of resistance determinants were detected [17]. Another study concerning antibiotic resistance in *Aeromonas* spp. isolated from retail seafood revealed that 98% of the tested isolates were resistant to three or more antibiotics [18]. However, in general, the occurrence of antimicrobial-resistant bacteria in seafood isolates decreases with increasing distance from possible sources of fecal contamination [19].

*Pseudomonas* is the dominant bacterial genus in food processing facilities and is a common part of the microflora of many different food products [17,20,21,22]. The only species of *Pseudomonas* that is considered a human pathogen is *P. aeruginosa*, but this species is not a common part of the microflora in food products [17]. Additionally, *P. putida* has been reported as an opportunistic human pathogen [23,24]. *Pseudomonas* spp. have been reported as major spoilage bacteria in aerobically stored chilled fish and in processing equipment, and *P. fluorescens*, *P. lundensis*, *P. libanensis*, *P. gessardii* and *P. veronii* have been detected in salmon filet and processing equipment [20].

The aim of this study was to classify *Pseudomonas* isolates and examine the occurrence of antibiotic resistance in these isolates and to estimate a possible AMR contribution by this genus in the salmon processing plant. This was performed phenotypically by the disk diffusion method and genotypically by whole genome sequencing. The isolates were collected from a Norwegian salmon processing plant during the first year the processing plant was operative.

## 2. Materials and Methods

### 2.1. Sampling and Preparation of Isolates

The sampling was carried out in a newly opened salmon processing facility on the coast of Mid-Norway, which receives salmon from several marine farming locations in the region. The fish is pumped into the facility directly from the well boat without the use of waiting pens and the products made at the facility are gutted whole fish, whole fillets (with or without skin), and vacuum-packed portioned fillets with or without skin. Samples for this study were collected at four different time points throughout the first year of production and from seven sampling points, both food contact surfaces and non-food contact surfaces (Table 1). Additionally, samples of fish fillet and swab samples of skin and gills from gutted whole fish were collected. All surface samples were taken after cleaning and disinfection. Sampling was performed by swabbing 100 cm^2^ with a sterile swab (Promedia ST-25 PBS, r-biopharm, Pfungstadt, Germany) in 10 mL phosphate buffered saline (PBS) or by swabbing 900 cm^2^ (30 × 30 cm) with sterile cloths pre-moistened with 25 mL PBS (Sodibox, Névez, France). The choice of swabbing method depended on the type and area of the surface. All samples were kept cold during the transportation (3 h) from the facility to the lab. Dilutions and plating were performed the same day. Swab samples were rigorously vortexed and 10-fold serial dilutions were prepared before plating on *Pseudomonas* CFC Selective agar (CM0559 and SR0103, Oxoid Ltd., Basingstoke, UK). Cloth samples were supplemented with PBS to a final weight of 50 g more than a new unused cloth and mashed in a Stomacher for 30 s. Ten-fold serial dilutions were prepared before plating. From the fish fillet samples, 25 g were added to a Stomacher bag and PBS was added to a total weight of 250 g, mashed in a Stomacher for 30 s, 10-fold serial diluted and plated. Incubation conditions for samples on *Pseudomonas* CFC selective agar plates were 25 °C for 48 h. After quantification, single colonies were picked and re-propagated twice as a minimum before they were transferred to TSB w/20% glycerol and stored at −80 °C. 

### 2.2. Classification of Presumptive Pseudomonas Isolates by Sequencing of rpoD Gene or 16S rRNA Gene

A total of 222 presumptive *Pseudomonas* spp. isolates from selected sampling points were subjected to *rpoD* or 16S rDNA sequencing (Appendix A). DNA extractions were carried out by DNeasy Blood & Tissue Kit (Qiagen, Hilden, Germany) and performed according to the manufacturer (Dneasy Blood & Tissue Handbook, July 2006).

As all of these isolates were isolated from *Pseudomonas* CFC Selective agar, they were considered presumptive *Pseudomonas* spp. and were subjected to PCR for the *rpoD* housekeeping gene with primers PsEG30F (5′-ATYGAAATCGCCAARCG-3′) and PsEG790R (5′-CGGTTGATKTCCTTGA-3′), resulting in a 760 bp product [25]. The PCR reactions were performed with 25 μL reactions containing 1× PCR buffer, 200 μM of each nucleotide, total concentration of MgCl_2_ of 650 µM, 0.5 μM of each primer, 2.5 U Taq polymerase (Qiagen), and 50–100 ng template DNA. The PCR amplification cycles were as follows: initial denaturation at 95 °C for 15 min, 30 cycles of denaturation at 95 °C for 60 s, annealing for 60 s at 55 °C, and extension at 72 °C for 60 s, followed by a final extension at 72 °C for 5 min. Visualization of the PCR products was done on a 1% agarose gel. As the primers should be specific for the genus *Pseudomonas*, an amplicon of incorrect size, or a missing band of the correct size, was considered as an indication for the isolate being non-*Pseudomonas*. These isolates were subjected to PCR with the universal 16S primers 338F (5′-ACTCCTACGGGAGGCAGCAG-3′) [26] and 1492R (5′-GGTTACCTTGTTACGACTT-3′) [27], resulting in an amplicon of 1154 bp and covering V3-V9 variable regions. The PCR reactions were performed with 25 μL reactions containing 1× PCR buffer, 200 μM of each nucleotide, total concentration of MgCl_2_ of 650 µM, 0.4 μM of each primer, 2.5 U Taq polymerase (Qiagen), and 50–100 ng template DNA. The PCR amplification cycles were as follows: initial denaturation at 95 °C for 15 min, 30 cycles of denaturation at 95 °C for 60 s, annealing for 30 s at 58 °C, and extension at 72 °C for 60 s, followed by a final extension at 72 °C for 5 min. 

The PCR products were enzymatically purified by ExoSAP-IT™ (Thermo Fischer Scientific, Waltham, MA, USA) procedure, which entailed incubation at 37 °C for 15 min to degrade remaining primers and nucleotides, followed by inactivation at 80 °C for 15 min. Purified PCR products were quality controlled and prepared for sequencing according to Eurofins LightRun sequencing requirements. Classification of sequences was done by using BLASTn and comparison to sequences currently available in the NCBI database (www.ncbi.nlm.nih.gov/BLAST, last accessed on 16 March 2021).

A phylogenetic tree was constructed of selected *rpoD* sequences by using Geneious Prime v2022.1.1 (Dotmatics, Boston, MA, USA). The trimmed sequences were aligned and trimmed to equal length (~700 bp) and used to construct a phylogenetic tree (neighbor joining tree with Jukes–Cantor distance measure and bootstrap (100 replicates)). The constructed tree was exported to iTol and processed for better visualization [28].

### 2.3. Antibiotic Susceptibility by Disc Diffusion Assay 

The antibacterial susceptibility profiles of isolates from *Pseudomonas* CFC agar were performed using the conventional disk diffusion assay on 16 different antimicrobials from seven different classes. The selection of antibiotics for this screening was mainly based on the most used antibiotics in Norway in both human and veterinary medicine and the two antibiotics florfenicol and oxolinic acid, which are most used in aquaculture according to [8]. The method was conducted in accordance with the guidelines from the European Committee on Antimicrobial Susceptibility Testing [29] with modifications regarding incubation temperature and time. A 0.5 McFarland standard suspension was used for inoculum standardization of all isolates and the reference cultures *Pseudomonas aeruginosa* CCUG 17619 and *E. coli* CCUG 17620 were included. A few isolates that did not grow on conventional Mueller–Hinton agar plates (Oxoid, CM0337B, Basingstoke, UK)) were grown on Mueller–Hinton with sheep blood (Labolytic, 634-0676, Trondheim, Norway) containing the following antibacterial agents: ampicillin (AMP, 10 µg), amoxicillin (AML, 30 µg), piperacillin/tazobactam (TZP, 36 µg), piperacillin (PRL, 30 µg), cefotaxime (CTX, 30 µg), ceftriaxone (CRO, 30 µg), ceftazidime (CAZ, 30 µg), meropenem (MEM, 10 µg), imipenem (IPM, 10 µg), ciprofloxacin (CIP, 1 µg), amikacin (AK, 30 µg), tobramycin (TOB, 30 µg), doxycycline (DO, 30 µg), tetracycline (TET, 30 µg), oxolinic acid (OA, 2 µg) and florfenicol (FFC, 30 µg). Plates were incubated at 25 ± 2 °C for 20 h. Zones of inhibition were interpreted in accordance with the EUCAST breakpoint table [30]. For organisms where no guidelines exist, interpretative criteria for similar antimicrobial or organism combinations were used [31,32]. Multidrug resistant (MDR) strains in this study were defined as being resistant to antibiotics in three or more of the antimicrobial classes analyzed [33].

### 2.4. Whole Genome Sequencing

Thirty *Pseudomonas* isolates were selected, based on phenotypic resistance to antibiotics of four or more classes, for further characterization by whole genome sequencing (WGS). High quality DNA was extracted by using the Genomic Micro AX Bacteria+ Gravity-kit (102–100 M, A&A BIOTECHNOLOGY, Gdańsk, Poland) according to the manufacturer’s procedure. RNAse treatment was included in the procedure. The quality of the DNA was checked on agarose gel and DNA concentrations were estimated by spectrophotometric measurement using BioTek PowerWave XS (Winooski, VT, USA), Take3 plate and Gen5 2.0 software (BioTek Instruments Inc., Winooski, VT, USA). DNA samples were sent on ice with overnight shipment to Novogene UK Sequencing laboratory. DNA purity and integrity was again controlled, and accurate DNA concentration was measured by Qubit^®^ 3.0 fluorometer quantification at the sequencing laboratory. The genomic DNA was randomly sheared into short fragments, then end-repaired and A-tailed before Illumina adapters were ligated. A PCR amplification of the fragments with adapters was performed before size selection and purification. The sequencing strategy was paired-end sequenced with a read length of 150 bp at each end, performed on the Illumina^®^ NovaSeq^TM^ 6000 sequencing platform.

Base calling was done with CASAVA v1.8 software and the raw read dataset was subject to quality filtering. Paired reads containing either adapter contamination, more than 10% uncertain nucleotides or reads with low quality nucleotides (base quality Q ≥ 5) constituting more than 50% of either read, were removed to obtain high quality reads.

### 2.5. Data Analysis of Sequences

The whole genome sequences were analyzed by using the online web-based tools developed by the Center for Genomic Epidemiology (CGE). The high-quality read files were used as templates and uploaded to the typing tool KmerFinder 3.2 [34,35,36] to identify the species based on Kmers (length = 16 bases). The high-quality clean reads (fastq) were then assembled in Geneious Prime 2022.1.1 (by mapping to respective reference genomes (Appendix A). The read sets were paired during import to Geneious by using Bbmerge, Paired end (inward pointing) with insert size: 350 bp. Normalization on the sequence reads was performed by BBNorm v.38.84 with default settings: target coverage level = 40, min depth = 6 and no error correction. Assemblies for each isolate were generated by mapping to suitable references according to previous analysis. Geneious mapper was used with the settings: medium-low sensitivity and iteration up to five times. The consensus sequences were extracted to fasta files with the lowest stringency to get the fewest ambiguous bases.

To analyze the isolates for antimicrobial resistance determinants, both the ResFinder 4.1 webtool [36,37,38] and the Resistance Gene Identifier (RGI) in The Comprehensive Antimicrobial Resistance Database (CARD) [39] were used. The high quality read files were uploaded to ResFinder 4.1 and default settings ((threshold for ID = 90%, Min. length = 60%), all antimicrobial configurations and species = Other) were used. The assembled genomes for the isolates were uploaded to the RGI tool in CARD with settings: perfect, strict and loose hits and, nudge ≥ 95% identity loose hits to strict. Low sequence quality was selected to account for possible mistakes in the assemblies.

A phylogenetic tree of the 30 isolates’ genome assemblies and relevant reference genomes downloaded from GenBank was generated with Fast mode in the webtool NDtree 1.2 [40,41,42]. The newick file based on the UPGMA algorithm from NDtree was uploaded to iTol [28] for better visualization, and the tree was rooted at the *P. aeruginosa* outgroup.

Pairwise Average Nucleotide Identity (ANI) values between the 30 isolates’ genome assemblies and ten reference genomes was calculated using CJ bioscience’s online ANI Calculator from ChunLab [43], which is based on the OrthoANIu algorithm.

The raw read sequencing data are deposited in the NCBI Short Read Archive (SRA) under BioProject ID PRJNA856124.

## 3. Results

### 3.1. Species Relation and Diversity on Pseudomonas CFC Agar

Presumptive *Pseudomonas* isolates (*n* = 222) were collected during the first year of production of a salmon processing plant from the processing equipment and salmon skin, gills, and fillet. The isolates were identified by either sequencing of the 16S rRNA gene (*n* = 95) or the *rpoD* gene (*n* = 127) (Appendix A). The majority of these isolates (68%) were classified within the genus *Pseudomonas*, other identified genera were *Aeromonas*, *Acinetobacter*, *Morganella*, *Serratia*, *Shewanella*, *Stenotrophomonas*, and *Pseudoalteromonas* (Appendix A). Four isolates were classified only to family level and as members of the family *Enterobacteriaceae*. Within the *Pseudomonas* genus, 23 different species were detected, with *P. fluorescens* being the most abundant (42%; Appendix A). Other abundant *Pseudomonas* spp. classified were *P. azotoformans*, *P. gessardii*, *P. libanensis*, *P. lundensis*, *P. cedrina and P. extremaustralis*, which all belong to the *P. fluorescens* group [44]. In total, 27% of the *Pseudomonas* isolates could not be classified to species level.

A phylogenetic tree was constructed of *rpoD* sequences longer than 650 bp (Figure 1). A large group of the isolates clustered close to *P. fluorescens* and clustered to species within the *P. fluorescens* group. Isolates from different sampling points and different sampling dates were broadly distributed across the whole phylogenetic tree, e.g., LJP374 from sampling 2, eight months after the startup of the facility, and LJP883 from the fish, skin and gills sampled 12 months after startup, seem to be closely related. One isolate sampled after eight months from inlet water, LJP343, had a high similarity to an isolate from the slaughter department, LJP760, sampled after 12 months.

Strains with highest similarity to *P. fluorescens* were detected at all sampling dates and all sampling points in this study, *P. lundensis*, *P. gessardii*, *P. cedrina* and *P. azotoformans* were detected at multiple sampling dates and sampling points (Appendix A), while *P. extremaustralis* was only detected on the salmon’s skin and *P. anguilliseptica* only on the gills.

### 3.2. Phenotypic Antimicrobial Susceptibility

A total of 16 different antibiotics belonging to seven different classes of antibiotics were included in the susceptibility testing of the 222 isolates. Seven of the isolates (three *Pseudoalteromonas*, one *Stenotrophomonas*, and three *Pseudomonas*) did not meet the criteria of growth on Mueller–Hinton agar plates required in the guidelines, and susceptibility could not be determined. The non-*Pseudomonas* isolates were mainly resistant to ampicillin and amoxicillin. However, resistance to 12 out of 16 antibiotics was detected among these isolates, of which *Serratia* spp. and *Stenotrophomonas* spp. showed the highest resistance levels to the antibiotics tested (Appendix A).

A large proportion of the *Pseudomonas* isolates, 92% and 87% respectively, were resistant to ampicillin and amoxicillin (Figure 2). High levels of resistance towards oxolinic acid (92%) and florfenicol (84%) were also detected. Furthermore, resistance to the cephalosporins, cefotaxime and ceftriaxone were observed in 56% and 40% of the *Pseudomonas* isolates, while resistance to ciprofloxacin was observed in 9.5% (Figure 2; Appendix A). Among these *Pseudomonas* isolates, no resistance to amikacin or tobramycin was observed.

Most of the *Pseudomonas* isolates (86%) from the first sampling date were multidrug resistant (MDR). These isolates were detected at three sampling points: the conveyor belt (CSL) in the slaughter department, the suction unit of the gutting machine (G) and the inlet water (IW). These include the only isolate resistant to six different antibiotic classes, which was identified as *P. fluorescens* (LJP028) by *rpoD* sequencing (Table 2 and Appendix A). Additionally, this was the only isolate resistant to doxycycline. On the second sampling date, isolates were retrieved from five sampling points (Table 2). Isolates from the inlet water (IW) were mainly resistant to one or more of the following antibiotics: ampicillin, amoxicillin, oxolinic acid, and florfenicol. One of these isolates, *P. fluorescens* (LJP316), was also resistant to ciprofloxacin. From the third sampling date, the isolates with resistance to most antibiotics were *P. gessardii* (LJP706 and LJP707) isolated from the drain in the slaughter department (DS). These were the only isolates resistant to piperacillin/tazobactam in this study (Table 2 and Appendix A).

The resistance properties of isolates from fish fillet, skin and gills were highly diverse. Isolates resistant to less than three antibiotic classes accounted for 25%. In this group, isolates with resistance to most antibiotics were LJP844 (*P**. azotoformans*), LJP889 (*P. fluorescens*) and LJP888 (*P. libanensis*), isolated from the fish skin. Two of the three *Pseudomonas* isolates susceptible to all tested antibiotics in this study were detected in this group.

### 3.3. Genomic Characterization Based on WGS Data

The typing tool KmerFinder 3.2 provided a classification for the strain most similar to each isolate (Appendix A) together with a score that gives the total number of matching Kmers between the query and the template, and Query Coverage (%) and Template Coverage (%), which gives the percentage of input query Kmers that match the template and the template coverage respectively. Depth gives an estimation of the sequencing depth. For seven of the isolates, the best match from KmerFinder obtained low values for Kmer match between query sequence and template sequence, which indicates low similarity to any other genome in the database.

The phylogeny of the 30 *Pseudomonas* isolates subjected to WGS is visualized in Figure 3. The tree shows two main groups, one small cluster with the three isolates LJP316, LJP321, LJP379 and the reference strain *Pseudomonas* sp. NIBR-H-19 and one large cluster where the rest of the isolates and reference strains are in smaller subgroups. The latter includes species like *P. sivasensis*, *P. gessardii*, *P. fluorescens*, *P. synxantha*, *P. libanensis* and the unclassified *Pseudomans* sp. FDAARGOS_380 and *Pseudomonas* sp. J380. An overview of all reference genes and genomes used in this study can be found in Table 3.

Eight of the isolates (LJP026, LJP028, LJP031, LJP039, LJP040, LJP043, LJP044, LJP045) clustering together with *Pseudomonas* sp. J380 originated from the first sampling but from two different sampling points: the conveyor and the gutting machine in the slaughter department of the facility. Two other isolates (LJP418 and LJP426) highly similar to the eight, originated from the second sampling and the head cutter, which is downstream of the previously mentioned sampling point. Five additional isolates from samplings two and three are closely related to this group. These were detected in the gutting machine and the head cutter during sampling two, and on a conveyor in the slaughter department and on fish skin during sampling three.

The pairwise calculated ANI values (Appendix A) support the clusters in the phylogenetic tree. By using the ANI value cutoff at ≥99.00% for strains and ≥96.50% for species [45,46], these pairwise ANI calculations between the isolates and ten reference genomes revealed 11 different species among our isolates as indicated by the colored boxes in Figure 3.

The 30 isolates that were subjected to WGS were selected based on phenotypic resistance to four or more classes of antibiotics. None of these isolates carried any acquired antibiotic resistance genes according to the ResFinder 4.1 database. However, according to CARD RGI strict hits, three different antimicrobial resistance determinants were present among the isolates (Table 3). These were *adeF*, the *Pseudomonas aeruginosa soxR* (*Paer_soxR*), and *Acinetobacter baumanii AbaQ* (*Abau_AbaQ*). While *Abau_AbaQ* is a MFS transporter that directly pumps fluoroquinolone antibiotics out of the cell to confer resistance, both *adeF* and *Paer_soxR* are parts of efflux pump complexes that confer antibiotic resistance. Additionally, all isolates had more than 400 loose hits (sequences with a match bitscore less than the curated BLASTP bitscore) on antibiotic determinants registered in CARD RGI, including several different genes known to be involved in various multidrug efflux pump systems in *P. aeruginosa* (Appendix A).

## 4. Discussion

*Pseudomonas* spp. are recognized as major food spoilers in the food industry, in salmon processing plants [20], in poultry [17], and in the dairy and meat industry [22].

The origin of the material in this study was colonies grown on *Pseudomonas* CFC Selective agar after sampling in a salmon processing facility over a period of one year. Among the *Pseudomonas* isolates analyzed in this study, 23 different species were detected, with *P. fluorescens* being the most abundant. Other abundant species were *P. azotoformans*, *P. gessardii*, *P. libanesis*, *P. lundensis*, *P. cedrina* and *P. extremaustralis*, which all belong to the *P. fluorescens* lineage according to Girard et al. [45], albeit different groups and subgroups. However, as many as 32% of the isolates detected from *Pseudomonas* CFC Selective agar belonged to genera other than *Pseudomonas*. These were classified as species of *Aeromonas*, *Acinetobacter*, *Morganella*, *Serratia*, *Shewanella*, *Stenotrophomonas*, and *Pseudoalteromonas*. It is known that bacteria from other genera are able to grow on *Pseudomonas* CFC Selective agar [17,47]. As many of the non-*Pseudomonas* species in this study are potential food spoilage bacteria, they were included in the further analyses and served as a basis of comparison in the antibiotic susceptibility tests.

Even though the reports of *Pseudomonas* spp. in various environments are frequent, the reported species vary. For example, in the salmon industry, reported *Pseudomonas* species are *P. fluorescens*, *P. lundensis*, *P. libanensis*, *P. gessardii* and *P. veronii* [20]. In meat and dairy environments, *P. fragi* and *P. fluorescens* were found to be most prevalent [22], while Heir et al. [17] reported species of the *P. fluorescens* lineage (*P. gessardii*, *P. lactis*, *P. weihenstephanensis*) to be the most prevalent in chicken meat. As seen in our results, many of the isolates (27%) showed high similarity to various unclassified *Pseudomonas* spp., hence no species classification was achieved. Additionally, some isolates could not be differentiated between two or more known species due to equally high similarity to the different species. The genus *Pseudomonas* is large and complex with, at present, more than 300 validly described species [45] and additionally several hundred unclassified strains. For the genus *Pseudomonas*, sequencing of the 16S rRNA gene can in most cases only delineate the three main lineages (*P. aeruginosa*, *P. pertucinogena* and *P. fluorescens*) but cannot with confidence differentiate environmental isolates at the species level [44]. A MLST approach including the genes 16S rRNA, *rpoB*, *rpoD* and *gyrB* has been shown to provide a better resolution for *Pseudomonas* species identification [48]. Sequencing of only the *rpoD* has also been suggested, and proven, to be an accurate, inexpensive, and less laborious alternative for identification of large sets of environmental *Pseudomonas* isolates [25,46].

Among the 30 *Pseudomonas* isolates subjected to WGS, 21 had the highest similarity to different unclassified *Pseudomonas* spp. according to KmerFinder. Four isolates showed highest similarity to *P. fluorescens* (LJP030, LJP707) or *P. synxantha* (LJP374, LJP883), although the similarity was not very high. Compared to the *rpoD*-based classification, three of these isolates (LJP030, LJP374, LJP883) were most similar to *P. fluorescens* with a sequence identity above the cutoff limit of ≥98.0% as recommended by Girard et al. [46], while LJP707 was classified as *P. gessardii*. This discrepancy can be explained by database issues as the KmerFinder database contains only high-quality, complete, and annotated genomes, while the classification by *rpoD* was done by performing a BLASTn search in Genbank, which contains more than 40,000 registered *rpoD* sequences from various *Pseudomonas*.

Average Nucleotide Identity (ANI) is a widely used method to compare two bacterial genomes for classification. It is common to consider ANI values of ≥95.0% to indicate the species boundaries [49,50] but in the work with *Pseudomonas* species delineation, Girard et al. [45] set the cutoff at ≥96.5% to classify isolates to the same species and considered ANI values between 95.0% and 96.5% to be ambiguous.

Several of the isolates in this study were closely related to *Pseudomonas* sp. J380. Ten of the isolates must be considered the same strain and additionally, five are most likely of the same species, according to ANI values. Isolate LJP030, classified as *P. fluorescens* by *rpoD* sequencing (Id: 99.35%), has the highest similarity, though not so high, to *P. fluorescens* PF08 according to KmerFinder (Appendix A) and does not cluster with any other isolates or references in the phylogenetic tree. The isolate has the highest ANI value when compared to *P. fluorescens* PF08 at 93.32% and next, to *P. gessardii* (92.93%) and LJP707 (92.85%). Several of our analyses pointed towards this isolate belonging to *P. fluorescens* species, but the ANI values were below the cutoff for species delineation. However, the ANI values of the two *P. fluorescens* reference genomes ATCC 13525 and PF08 (84.82%) were also below this cutoff. It is not clear if this was caused by high heterogeneity within the *P. fluorescens* species or by mis-annotations in the database [51]. This issue, in addition to the fact that 21 of our 30 sequenced genomes are most similar to unclassified strains of *Pseudomonas* and, the high rate of unclassified species by *rpoD* sequencing, demonstrates the difficulties arising in *Pseudomonas* classification and shows that in many cases even WGS cannot determine the species identity with confidence.

A consequence of extensive use and misuse of antibiotics is the emergence of resistant bacteria. In aquaculture, only a small number of antibiotics are permitted to use. These include oxolinic acid and florfenicol, which are the most used antibiotics in both the Chilean and Norwegian aquaculture [8,9]. The use of these antibiotics is very low (223 kg) in the Norwegian aquaculture due to an efficient vaccination program in fingerlings [52]. However, in this study, we detected a high number of oxolinic acid and florfenicol-resistant bacteria. Additionally, a few isolates were susceptible to ampicillin and amoxicillin. It was expected to see high levels of resistance towards these antibiotics as it is well documented that their relative, and human pathogen, *Pseudomonas aeruginosa*, displays resistance to several antibiotics from different classes, e.g., aminoglycosides, quinolones and the majority of related β-lactam antibiotics, e.g., ampicillin [53,54]. *Pseudomonas* spp. resistant to ampicillin have also been documented in other parts of the food industry [55,56]. However, among our environmental isolates, not all *Pseudomonas* were ampicillin resistant. Therefore, we found it relevant to also report these results. The level of resistance among isolates in this study did not increase with the time of sampling. This indicates that the selective pressure in the food processing environment did not induce increased resistance.

A large proportion of our *Pseudomonas* isolates (82%) were resistant to florfenicol. Similar results were described among *Pseudomonas* sampled close to a mussel farm and in shellfish in Chile [57,58]. Buschmann et al. [59] showed that florfenicol- and oxolinic acid-resistant bacteria could also be detected in the sediments beneath the fish cages and thereby increase the proportion of antimicrobial-resistant bacteria in the environment. One of the mechanisms associated with florfenicol resistance is the presence of the *floR* gene encoding florfenicol/chloramphenicol specific efflux pumps [10]. However, a study by Fernández-Alarcón et al. [60] found that florfenicol resistance does not necessarily correlate with the presence of the *floR* gene. In that study, florfenicol MIC values among Gram negative bacteria were determined in the presence and absence of specific efflux pump inhibitors. High MIC values were detected among bacteria both positive and negative for the *floR* gene. Further, Fernández-Alarcón et al. [60] pointed out that non-specific multi-drug efflux pump systems may be involved in resistance mechanisms. Likewise, Adesoji and Call [61] reported a high occurrence of florfenicol resistance among *Pseudomonas* spp. in combination with a low prevalence of the *floR* gene.

Several of the isolates in this study were highly similar to *Pseudomonas* sp. J380, which is described as an opportunistic pathogen in cunners (*Tautogolabrus adspersus*) and lumpfish (*Cyclopterus lumpus*), which are used as cleaner fish in salmon farming [62]. Most of the farmed fish and also the different cleaner fish species are susceptible to bacterial infections, which are commonly treated with florfenicol [63,64]. In this study, all the 15 isolates highly similar to *Pseudomonas* sp. J380 were resistant to florfenicol in the disc diffusion assay. This information should be of interest for the veterinary medicine society and taken into consideration when prescribing antibiotics to cleaner fish populations.

Resistance to other β-lactams was also observed in this study, 27% and 5% of the isolates were resistant to the carbapenems imipenem and meropenem, respectively. This is in line with results from *Pseudomonas* spp. from poultry where 26% and 13% were resistant to imipenem and meropenem [17]. In the dairy industry however, a variation in the resistance pattern in *Pseudomonas* spp. has been observed. *Pseudomonas* isolates from raw milk were highly resistant to imipenem (95%), and to a lower extent resistant to meropenem (28%) [55], while the opposite occurred in *Pseudomonas* isolates from bulk tank milk, highly resistant to meropenem (56%) and to a lesser extent to imipenem [56].

Despite the high level of phenotypic resistance, the search in the ResFinder database with sequence reads from WGS did not reveal any acquired resistance genes in these isolates. Hence, all phenotypic resistance observed in these isolates most likely originates from intrinsic mechanisms commonly found in pseudomonads and in particular described for *P. aeruginosa* [65,66,67]. For example, it has been observed that carbapenem resistance in *Pseudomonas* spp. is mostly mediated via efflux pumps, especially in aqueous environments [68]. *P. aeruginosa* is a well-studied human pathogen within the genus *Pseudomonas* and is known to inhabit high intrinsic resistance to several different antibiotics. The mechanisms behind such resistance can include a low outer membrane permeability, multidrug efflux pump systems such as MexAB-OprM or MexXY-OprM [69], and the production of inactivating enzymes like β-lactamases [70]. Some of these intrinsic resistance mechanisms can confer resistance to multiple antibiotics at once [71]. It can be reasonable to anticipate that the mechanisms causing resistance in other *Pseudomonas* spp. can be the same or similar to those described in *P. aeruginosa*. In the study by Heir et al. [17], genes encoding the MexAB-OprM efflux system were detected in 29 of 31 *Pseudomonas* strains of different species. The search in CARD RGI revealed that all of our 30 isolates carried the *adeF* gene, encoding the membrane fusion protein of the *adeFGH* multidrug efflux complex that can confer resistance to tetracyclines and fluoroquinolones, as described for *Acinetobacter baumanii* by Coyne et al. [72]. The *adeG* and *adeH* genes were not detected in any of the isolates among the strict hits in CARD. However, by also including loose hits, six of the isolates (LJP045, 341, 344, 426, 728 and 844) carried variants of both *adeG* and *adeH* genes, while the rest were lacking one or both. The *Pseudomonas aeruginosa soxR* (*Paer_soxR*) gene was detected in 27 of the isolates. This gene encodes a redox-sensitive transcriptional activator that induces expression of a small regulon that includes the RND efflux pump-encoding operon MexGHI-opmD [73] and by that, can confer resistance to several drugs including tetracycline, fluoroquinolone, penam, cephalosporin, glycylcycline, phenicol and rifamycin, in addition to disinfection agents and antiseptics. This system is described in *P. aeruginosa*. None of the isolates in this study carried all the genes in the MexGHI-opmD and it is therefore unlikely that this is the mechanism that confers the observed resistance in our isolates. However, all the isolates carried variants of all genes (loose hits) involved in several other multidrug efflux pump systems, which can confer the resistant properties that were observed in the disc diffusion assay (Appendix A). For example, eight isolates (LJP042, 045, 316, 321, 379, 719, 726 and 760) carry variants of all genes (loose hits) in the MexAB-OprM efflux system, which can confer resistance to all the antibiotic classes of which we observed resistance to. However, since these are loose hits, and since the isolates are not *P. aeruginosa*, we cannot conclude that these detected systems are the cause of the observed phenotypic resistance properties.

Twenty-six isolates carry the *Acinetobacter baumanii AbaQ* (*Abau_AbaQ*) gene, which encodes a major facilitator superfamily (MFS) transporter mainly involved in the extrusion of quinolone-type drugs in *A. baumanii* [74]. Variants of the gene also commonly occur in several *Pseudomonas* spp.

Antibiotic resistance was detected in a variety of species in this study; however, *P. gessardii* was the only species resistant to TZP. The findings of Heir et al. [17] indicated taxa specific differences in resistance properties. In our material, only three isolates were susceptible to all antibiotics tested. Of these, one was classified as *P. brenneri*, one as *P. anguilliseptica*, one as *P. fluorescens* and one unclassified. The *P. brenneri* and *P. anguilliseptica* were the only isolates of the respective species, and the *P. fluorescens* was one susceptible isolate among many resistant. Among the other species detected, variable resistance profiles were seen and there was no clear indication of taxa specific resistance profiles.

A prerequisite for using the inhibition zone interpretation criteria in the disc diffusion assay is the incubation of MHA (Müller–Hinton agar) plates at 35 ± 2 °C for 16–18 h. However, as most of the tested isolates were psychrophiles and could not grow at high temperatures, the assay for these was conducted at 25 °C for 20 h instead. Smith and Kronvall [75] demonstrated that the precision in sets of disc diffusion zones decreases with lower incubation temperature and increased time. The lower incubation temperature is probably the reason why we registered a slight deviation in the zone diameter for our reference strains for a few of the antibiotics (Appendix A). Similar issues when testing psychrophilic bacteria have been reported earlier [31,32]. Thus, it is clearly necessary to develop interpretive criteria allowing lower incubation temperatures to meet the need for resistance testing of aquatic isolates. The Clinical Laboratory Standard Institute (CLSI) have published standard test protocols for both disc diffusion assay and MIC methods for incubation at 28 ± 2 °C or 22 ± 2 °C, but this method and break-point tables were not accessible at the time of the experiment.

## 5. Conclusions

*Pseudomonas* isolates originating from a salmon processing environment are diverse with many species represented. But the complex and confusing taxonomy of the genus *Pseudomonas* makes it difficult to provide confident taxonomic assignments for many of the isolates. However, in this study, isolates belonging to the *P. fluorescens* group are highly dominating. The isolates show a high level of phenotypic resistance towards a panel of antibiotics with 86% of them being resistant towards three of more classes of antibiotics and hence must be considered as multidrug resistant. This resistance is most likely not caused by any acquired antimicrobial resistance gene, as no such genetic resistance determinants were detected in the set of 30 isolates subjected to whole genome sequencing. More likely it is caused by intrinsic stress response and/or efflux pump systems, which are known to be frequent among *Pseudomonas* spp. Variants of genes known from *P. aeruginosa* to be involved in such systems were detected in all the isolates. However, because of low sequence similarity to the described genes, further studies are needed to confirm their presence and function. As no acquired resistance genes were detected, the probability of spreading of the resistance to other bacteria within this food processing environment and further into the food value chain is small. However, the high level of phenotypic resistance is concerning and should be monitored. Finally, we would like to point out the finding of resistance to florfenicol in isolates with very high genomic similarity to *Pseudomonas* sp. J380, which was recently described as the cause of bacterial infections in different cleaner fish species.

## Figures and Tables

**Figure 1 microorganisms-10-01420-f001:**
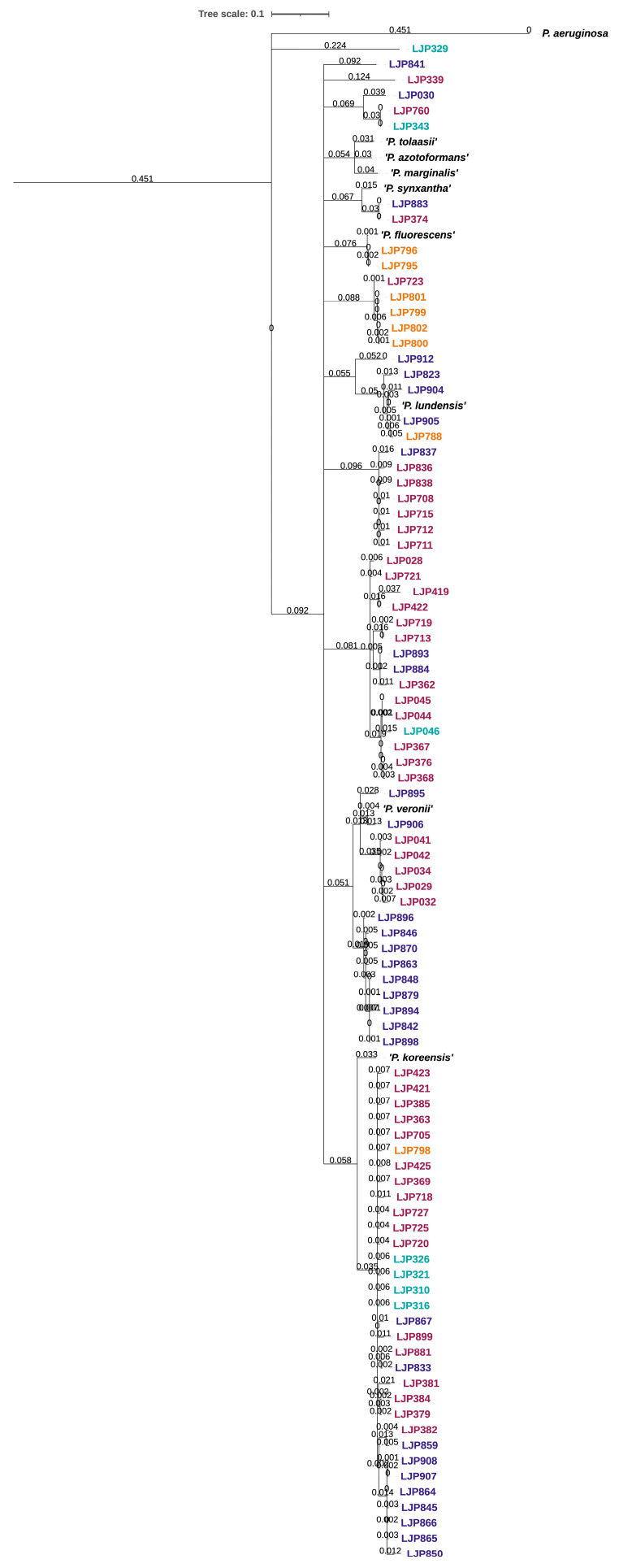
Phylogenetic distribution of *Pseudomonas* isolates (*n* = 89) based on partial sequencing of the *rpoD* gene. The trimmed sequences were aligned and cut to equal length (~700 bp) and used to construct a phylogenetic tree (neighbor joining tree with Jukes–Cantor distance measure and bootstrap (100 replicates)). The sampling point of isolation is indicated by color: inlet water (light blue), salmon slaughter department (dark pink), fillet department (orange), skin, gills and fish fillet (dark blue). The *rpoD* sequence of nine relevant reference strains are included and *P. aeruginosa* were used as an outgroup.

**Figure 2 microorganisms-10-01420-f002:**
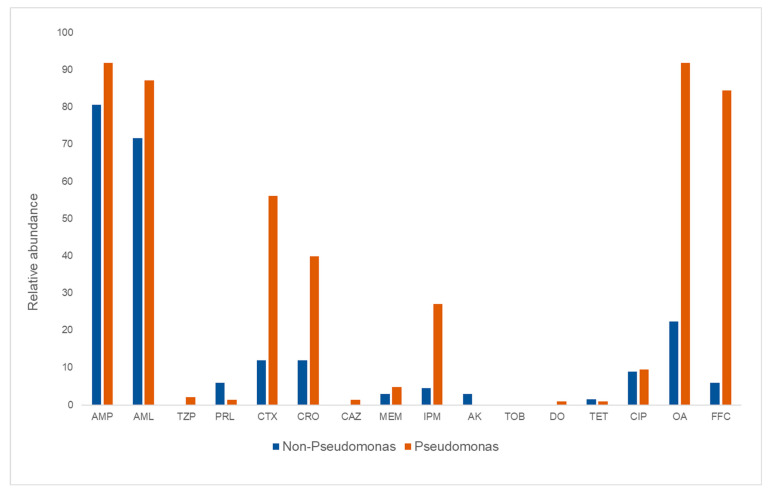
Relative abundance of *Pseudomonas* and non-*Pseudomonas* isolates sampled from different sampling points in a salmon processing facility harboring phenotypical resistance towards sixteen different antibiotics. AMP-ampicillin, AML-amoxicillin, TZP-piperacillin/tazobactam, PRL-piperacillin, CTX-cefotaxime, CRO-ceftriaxone, CAZ-ceftazidime, MEM-meropenem, IPM-imipenem, AK-amikacin, TOB-tobramycin, DO-doxycycline, TET-tetracycline, CIP-ciprofloxacin, OA-oxacilinic acid, FFC-florfenicol.

**Figure 3 microorganisms-10-01420-f003:**
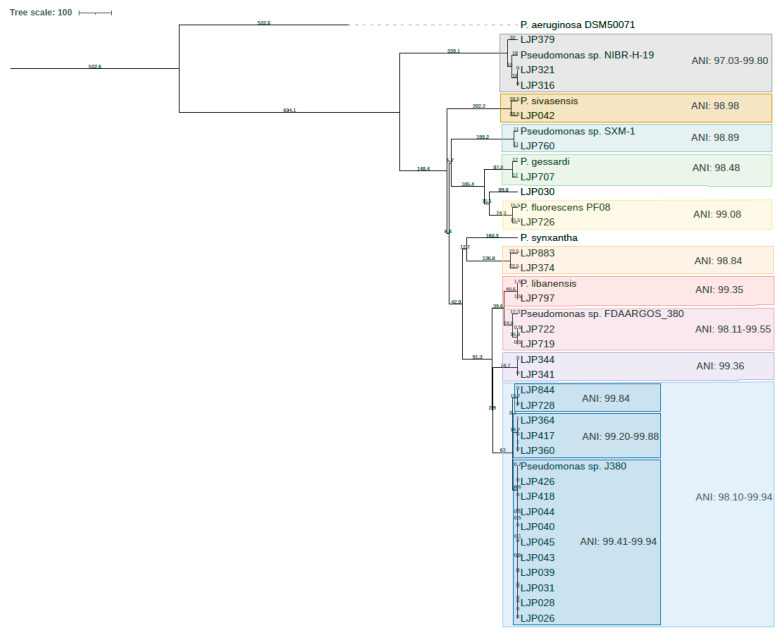
Phylogenetic tree based on draft genome assemblies of 30 environmental isolates of *Pseudomonas* spp. and ten reference genomes with *P. aeruginosa* as an outgroup. The UPGMA phylogenetic tree was generated by the online webtool NDTree and exported to iTOL for post processing. The different clusters are marked in different colors and the intra-group ANI values are included. The main groups here all have intra group ANI values ≥ 96.5% and are considered to belong to the same species. The large group highly similar to reference *Pseudomonas* sp. J380 is divided into three smaller clusters and the intra subgroup ANI values are noted. These intra subgroup ANI values are >99.0% and the isolates in each subgroup are considered to be the same strain.

**Table 1 microorganisms-10-01420-t001:** Overview of the different sampling points, sampling point category; CSS = contact surface slaughter department, NCS = non-contact surface, CSF = contact surface filleting department, F = fish, sampling type; water, cloth, swab or fish fillet, and approximate sampling area.

Sampling Point	Sampling Point Category	Sampling Type	Sampling Area
Inlet water	Contact surface, slaughter	Water	100 mL
Drain slaughter dep.	Non-contact	Cloth	30 cm × 30 cm
Conveyor slaughter dep.	Contact, slaughter	Cloth	30 cm × 30 cm
Gutting machine, suction	Contact, slaughter	Swab	10 cm × 10 cm
Head cutter knife	Contact, slaughter	Cloth	2 cm × Ø25 cm
Conveyor fillet dep.	Contact, fillet	Cloth	30 cm × 30 cm
Drain fillet dep.	Non-contact	Cloth	30 cm × 30 cm
Fish fillet before packaging	Fish	Fish fillet	25 g
Skin, gutted whole fish	Fish	Swab	10 cm × 10 cm
Gills, gutted whole fish	Fish	Swab	Gills on both side of fish

**Table 2 microorganisms-10-01420-t002:** Resistance profiles of the classified *Pseudomonas* population based on disk diffusion with corresponding breakpoint values (EUCAST 2022). All isolates are LJP, only isolate numbers are displayed in the columns. CSL; conveyor slaughter, G; gutting machine suction, IW; inlet water, HCK; head cutting knife, CSK; conveyor skinning, DS; drain slaughter, DF; drain filleting department, S1–5; skin five different fish, F1–5; Fillet five different fish, G1–5; gills of five different fish. The sampling times span a period of one year.

Sampl. Time	Sampl. Point	Isolate NO (LJP)	Taxonomic Classification	Resistance Profiles	Res. to # Antib. Classes
1	CSL	035, 040	*Pseuddomonas fluorescens* (2)	AMP, AML, CTX, CRO, IPM, OA, FFC	5
1	CSL	044	*Pseudomonas fluorescens*	AMP, AML, CTX, CRO, CAZ, IPM, OA, FFC	5
1	CSL	045	*Pseudomonas fluorescens*	AMP, AML, CTX, CRO, CIP, OA, FFC	4
1	CSL	042	*Pseudomonas fluorescens*	AMP, AML, CTX, CRO, OA, FFC	4
1	CSL	033, 038	*Pseudomonas tolaasii* (2)	AMP, AML, CTX, CRO, IPM, OA, FFC	5
1	CSL	037	*Pseudomonas umsongensis*	AMP, AML, CTX, CRO, IPM, OA, FFC	5
1	CSL	043	*Pseudomonas synxantha*	AMP, AML, CTX, CRO, IPM, CIP, OA, FFC	5
1	CSL	046	*Pseudomonas sp.*	AMP, AML, CTX, IPM, OA, FFC	5
1	CSL	032, 034, 041	Unclassified *Pseudomonas* (3)	AMP, AML, CTX, CRO, OA, FFC	4
1	CSL	039	*Pseudomonas cedrina*	AMP, AML, CTX, CRO, IPM, OA, FFC	5
1	G	028	*Pseudomonas fluorescens*	AMP, AML, CTX, CRO, IPM, CIP, DO, OA, FFC	6
1	G	030	*Pseudomonas fluorescens*	AMP, AML, CTX, CRO, MEM, IPM, OA, FFC	5
1	G	027	*Pseudomonas reactans*	AMP, AML, CTX, CRO, IPM, OA, FFC	5
1	G	029	Unclassified *Pseudomonas*	AMP, AML, IPM, OA, FFC	4
1	G	026	*Pseudomonas azotoformans*	AMP, AML, CTX, CRO, IPM, CIP, OA, FFC	5
1	G	031	*Pseudomonas cedrina*	AMP, AML, CTX, IPM, OA, FFC	5
1	IW	009	*Pseudomonas brenneri*	susceptible	0
2	IW	310, 326	*Pseudomonas fluorescens* (2)	AMP, AML, OA, FFC	3
2	IW	316	*Pseudomonas fluorescens*	AMP, AML, CIP, OA, FFC	3
2	IW	321	*Pseudomonas fluorescens*	AMP, AML, CTX, OA, FFC	4
2	IW	314, 315	*Pseudomonas guineae* (2)	OA	1
2	IW	309	*Pseudomonas marincola*	OA, FFC	2
2	IW	312	*Pseudomonas pseudoalcaligenes*	AMP	1
2	IW	313, 320	Unclassified *Pseudomonas* (2)	OA, FFC	2
2	IW	311	Unclassified *Pseudomonas*	AMP	1
2	IW	327, 329	Unclassified *Pseudomonas* (2)	OA	1
2	CSL	339	*Pseudomonas fluorescens*	AMP, AML, OA, FFC	3
2	G	366, 371	*Pseudomonas azotoformans* (2)	AMP, AML, IPM, OA, FFC	4
2	G	375	*Pseudomonas azotoformans*	AMP, AML, CTX, CRO, OA, FFC	4
2	G	360	*Pseudomonas cedrina*	AMP, AML, CTX, IPM, OA, FFC	5
2	G	362	*Pseudomonas fluorescens*	AMP, AML, CTX, CRO, OA, FFC	5
2	G	383	*Pseudomonas fluorescens*	AMP, AML, CTX, OA, FFC	4
2	G	374	*Pseudomonas fluorescens*	AMP, AML, CTX, CRO, CAZ, CIP, OA, FFC	4
2	G	364	*Pseudomonas fluorescens*	AMP, AML, CTX, CRO, IPM, OA, FFC	5
2	G	370	*Pseudomonas fluorescens*	AMP, AML, CTX, CRO, OA, FFC	4
2	G	373a	*Pseudomonas fluorescens*	AMP, AML, IPM, OA, FFC	4
2	G	369	*Pseudomonas fluorescens*	AMP, AML, OA, FFC	3
2	G	363, 381, 382, 384, 385	*Pseudomonas fluorescens* (5)	AMP, AML, OA, FFC	3
2	G	379	*Pseudomonas fluorescens*	AMP, AML, TET, OA, FFC	4
2	G	365, 372	Unclassified *Pseudomonas*	AMP, AML, OA, FFC	3
2	G	367	Unclassified *Pseudomonas*	AMP, AML, CTX, IPM, OA, FFC	5
2	G	368, 376	Unclassified *Pseudomonas* (2)	AMP, AML, CRO, IPM, OA, FFC	5
2	G	380	Unclassified *Pseudomonas*	AMP, AML, CRO, IPM, CIP, OA, FFC	5
2	HCK	421	*Pseudomoans lurida*	AMP, AML, CTX, CRO, OA, FFC	4
2	HCK	422	*Pseudomonas fluorescens*	AMP, AML, CTX, CRO, OA, FFC	4
2	HCK	419, 423, 425	*Pseudomonas fluorescens* (3)	AMP, AML, OA, FFC	3
2	HCK	417, 426	*Pseudomonas fluorescens* (2)	AMP, AML, CTX, CRO, IPM, CIP, OA, FFC	5
2	HCK	418	*Pseudomonas marginalis*	AMP, AML, CTX, CRO, IPM, CIP, OA, FFC	5
2	CSK	344	*Pseudomonas azotoformans*	AMP, AML, CTX, CRO, IPM, OA, FFC	5
2	CSK	341	*Pseudomonas fluorescens*	AMP, AML, CTX, CRO, IPM, OA, FFC	5
2	CSK	343	Unclassified *Pseudomonas*	AMP, AML, CRO, OA, FFC	4
3	DS	710	*Pseudomonas azotoformans*	AMP, AML, CTX, OA, FFC	4
3	DS	713	*Pseudomonas fluorescens*	AMP, AML, OA, FFC	3
3	DS	705	*Pseudomonas fluorescens*	AMP, AML, OA, FFC	3
3	DS	706	*Pseudomonas gessardii*	AMP, AML, TZP, CTX, CRO, MEM, OA, FFC	5
3	DS	707	*Pseudomonas gessardii*	AMP, AML, TZP, PRL, CTX, CRO, MEM, OA, FFC	5
3	DS	714	*Pseudomonas gessardii*	AMP, AML, CTX, CRO, OA, FFC	4
3	DS	716	*Pseudomonas gessardii*	AMP, AML, TZP, CTX, CRO, MEM, OA, FFC	5
3	DS	718	*Pseudomonas fluorescens*	AMP, AML, OA, FFC	3
3	DS	708, 711, 715	Unclassified *Pseudomonas* (3)	AMP, AML, CTX, OA, FFC	4
3	DS	712	Unclassified *Pseudomonas*	AMP, AML, CTX, OA, FFC	4
3	DS	709	Unclassified *Pseudomonas*	AMP, AML, OA, FFC	3
3	CSL	722	*Pseudomonas fluorescens*	AMP, AML, CTX, IPM, OA, FFC	5
3	CSL	721, 719	*Pseudomonas fluorescens* (2)	AMP, AML, CTX, CRO, IPM, OA, FFC	5
3	CSL	720, 727	*Pseudomonas fluorescens* (2)	AMP, AML, OA, FFC	3
3	CSL	725	*Pseudomonas fluorescens*	AMP, AML, OA, FFC	3
3	CSL	726	*Pseudomonas fluorescens*	AMP, AML, CTX, CRO, MEM, IPM, OA, FFC	5
3	CSL	728	*Pseudomonas paralactis*	AMP, AML, CTX, IPM, CIP, OA, FFC	5
3	CSL	724	*Pseudomonas poae*	AMP, AML, CTX, CRO, OA, FFC	4
3	CSL	723	Unclassified *Pseudomonas*	AMP, AML, CTX, CRO, MEM, IPM, OA, FFC	5
3	HCK	760	*Pseudomonas putida*	AMP, AML, CRO, OA, FFC	4
3	CSK	788	*Pseudomonas lundensis*	AMP	1
3	DF	796	*Pseudomonas fluorescens*	AMP, AML, CTX, CRO, OA, FFC	4
3	DF	799, 800, 801	Unclassified *Pseudomonas* (3)	AMP, AML, CTX, CRO, OA, FFC	4
3	DF	802	Unclassified *Pseudomonas*	AMP, AML, CTX, OA, FFC	4
3	DF	795	*Pseudomonas fluorescens*	AMP, AML, CTX, CRO, OA, FFC	4
3	DF	798	*Pseudomonas fluorescens*	AMP, AML, OA, FFC	3
3	DF	797	*Pseudomonas azotoformans*	AMP, AML, CTX, CRO, IPM, OA, FFC	5
3	G2	823	*Pseudomonas anguilliseptica*	susceptible	0
3	S1	844	*Pseudomonas azotoformans*	AMP, AML, CTX, CRO, IPM, CIP, OA, FFC	5
3	S1	840	*Pseudomonas extremaustralis*	AMP, AML, CTX, OA, FFC	4
3	S5	910	*Pseudomonas extremaustralis*	AMP, AML, CTX, CRO, OA, FFC	4
3	S1	843	*Pseudomonas extremaustralis*	AMP, AML, CTX, OA	3
3	S5	899	*Pseudomonas fluorescens*	AMP, AML, OA, FFC	3
3	S1	836	*Pseudomonas fluorescens*	AMP, AML, OA, FFC	3
3	S3	867	*Pseudomonas fluorescens*	AMP, AML, CIP, OA, FFC	3
3	S4	880	*Pseudomonas fluorescens*	AMP, AML, CTX, OA, FFC	4
3	S4	881	*Pseudomonas fluorescens*	AMP, AML, OA, FFC	3
3	S4	883	*Pseudomonas fluorescens*	AMP, AML, CTX, CRO, IPM, CIP, OA, FFC	5
3	S4	889	*Pseudomonas fluorescens*	AMP, AML, CTX, CRO, MEM, IPM, OA, FFC	5
3	S5	906	*Pseudomonas fluorescens*	AMP, AML, CTX, CRO, OA, FFC	4
3	S5	907, 908	*Pseudomonas fluorescens* (2)	AMP, AML, OA	2
3	S1	845	*Pseudomonas fluorescens*	AMP, AML, OA, FFC	3
3	S2	850	*Pseudomonas fluorescens*	AMP	1
3	S2	859	*Pseudomonas fluorescens*	susceptible	0
3	S3	864	*Pseudomonas fluorescens*	AMP, AML, OA	2
3	S3	865	*Pseudomonas fluorescens*	AMP, AML	1
3	S3	866	*Pseudomonas fluorescens*	AML	1
3	S4	887	*Pseudomonas fluorescens*	AMP	1
3	S5	912	*Pseudomonas fragi*	AMP, AML, PRL, CTX, OA, FFC	4
3	S1	835	*Pseudomonas gessardii*	AMP, AML, CTX, OA, FFC	4
3	S4	893	*Pseudomonas libanensis/fluorescens*	AMP, AML, OA	2
3	S4	882	*Pseudomonas libanesis*	AMP, AML, OA, FFC	3
3	S4	888	*Pseudomonas libanesis*	AMP, AML, CTX, CRO, IPM, OA, FFC	5
3	F2	833	*Pseudomonas lundensis*	AMP, OA, FFC	3
3	S5	905	*Pseudomonas lundensis*	AMP, OA, FFC	3
3	S5	904	*Pseudomonas lundensis*	AMP	1
3	S5	895	*Pseudomonas veronii*	AMP, AML, CTX, CRO, OA, FFC	4
3	S4	884	*Pseudomonas libanensis*	AMP, AML, CTX, OA, FFC	4
3	S5	896	Unclassified *Pseudomonas*	AMP, AML, CTX, CRO, OA, FFC	4
3	S1	846	Unclassified *Pseudomonas*	AMP, AML, CTX, CRO, OA, FFC	4
3	S3	863, 870	Unclassified *Pseudomonas* (2)	AMP, AML, CTX, CRO, OA, FFC	4
3	S1	838, 839, 842, 837	Unclassified *Pseudomonas* (4)	AMP, AML, CTX, OA, FFC	4
3	S1	841	Unclassified *Pseudomonas*	AMP, AML, OA, FFC	3
3	S2	848	Unclassified *Pseudomonas*	AMP, AML, CTX, OA, FFC	4
3	S4	879	Unclassified *Pseudomonas*	AMP, AML, OA, FFC	3
3	S5	894	Unclassified *Pseudomonas*	susceptible	0
3	S5	898, 903	Unclassified *Pseudomonas* (2)	AMP, AML, CTX, OA	3

**Table 3 microorganisms-10-01420-t003:** Detected resistance determinants in 30 *Pseudomonas* isolates and the associated predicted resistance properties.

Isolates (LJP)	Antimicrobial Resistance Determinants	Predicted Antimicrobial Resistance
726, 030, 707	*adeF*	fluoroquinolone; tetracycline
418	*adeF*, *soxR*	fluoroquinolone; cephalosporin; glycylcycline; penam; tetracycline; rifamycin; phenicol; disinfecting agents and antiseptics
026, 028, 031, 039, 040, 042, 043, 044, 045, 316, 321, 341, 344, 360, 364, 374, 379, 417, 426, 719, 722, 728, 760, 797, 844, 883	*adeF*, *soxR*, *AbaQ*	

## Data Availability

The raw read sequencing data are deposited in the NCBI Short Read Archive (SRA) and are available under BioProject ID PRJNA856124.

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
