# Peer review of "Antibiotic Resistance Properties among Pseudomonas spp. Associated with Salmon Processing Environments"

_microorganisms, 2022, doi:10.3390/microorganisms10071420_

Round 1
Reviewer 1 Report
The authors present an interesting investigation as they characterize well the contamination by Pseudomonas spp. associated with salmon processing environments and its antimicrobial resistance potential.
A few formatting errors should be corrected and a few questions answered:
Line 221-222: not in bold
3.3 Subtitle ?
Table 2: If we have counted correctly in table 2 there are 165 of us, is this correct?. Are we to understand that no Pseudomonas could be isolated from gills?
Line 514: Twenty-six or twenty six
References: should be revised as many errors need to be corrected in order to unify the format (i.e.: names of journals choose only one :in abbreviated or unabbreviated form; doi reference choose one: https:// or not; (try to search for all the doi references) comma or not before the year, etc.). Align text.
Author Response
The authors are grateful for constructive and valuable comments on our manuscript. We have considered all comments carefully and revised our manuscript accordingly.
Line 221-222: not in bold
A: The line is now not in bold. NCBI SRA BioProject ID is added.
3.3 Subtitle ?
A: It was only the number that was wrong. 3.4 is corrected to 3.3. (line 309)
Table 2: If we have counted correctly in table 2 there are 165 of us, is this correct?. Are we to understand that no Pseudomonas could be isolated from gills?
A: All the numbers and isolated related to table 2 are checked and double-checked. We identified 151 Pseudomonas, three of these did not grow on Müeller-Hinton agar, hence were not tested for antibiotic susceptibility. A total of 148 Pseudomonas isolates were tested for antibiotic susceptibility and are included in table 2. Only one Pseudomonas isolate was recovered from gills (LJP823, identified as P. anguilliseptica) and this isolate was susceptible to all antibiotics tested. Samples from gills are now mentioned in the legend for table 2 for clarification.
Line 514: Twenty-six or twenty six
A: This is corrected to Twenty-six.
References: should be revised as many errors need to be corrected in order to unify the format (i.e.: names of journals choose only one :in abbreviated or unabbreviated form; doi reference choose one: https:// or not; (try to search for all the doi references) comma or not before the year, etc.). Align text.
A: All references has been thoroughly revised and should now be in a unified format.
Reviewer 2 Report
The topic of the article falls within the thematic scope of the journal MICROORGANISMS.
The purpose of this study was to classify Pseudomonas isolates obtained from a Norwegian salmon processing plant, investigate their resistance to antibiotics, and evaluate the possible contribution of this type to antimicrobial resistance in a fish processing plant. This was performed using the disk diffusion method and by whole genome sequencing. I have no comments regarding the methods used and the interpretation of the results obtained.
The manuscript is prepared very carefully and does not require any changes.
I have only few comments to the manuscript (in section References) - all suggestions for corrections were introduced in the review mode to the attached pdf file.
In my opinion, the manuscript may be submitted to further editorial work.

Author Response
The manuscript is prepared very carefully and does not require any changes.
I have only few comments to the manuscript (in section References) - all suggestions for corrections were introduced in the review mode to the attached pdf file.
A: All references has been thoroughly revised and should now be in a unified format.
In my opinion, the manuscript may be submitted to further editorial work.